# Characterisation of *ACP5* missense mutations encoding tartrate-resistant acid phosphatase associated with spondyloenchondrodysplasia

Janani Ramesh[1], Latha K. Parthasarathy[2], Anthony J. Janckila[3], Farhana Begum[1], Ramya Murugan[1], Balakumar P. S. S. Murthy[4], Rif S. El-Mallakh[2], Ranga N. Parthasarathy[1,2,5], Bhuvarahamurthy Venugopal[1]*

1 Department of Medical Biochemistry, Dr. ALM-PGIBMS, University of Madras, Madras, India, 2 Department of Psychiatry, University of Louisville School of Medicine, Louisville, KY, United States of America, 3 Department of Microbiology and Immunology, University of Louisville, School of Medicine, Louisville, KY, United States of America, 4 Department of Vascular and Endovascular Sciences, Tamilnadu Government Multi Super Speciality Hospital, Chennai, India, 5 Department of Psychiatry, Molecular Biology and Biochemistry, University of Louisville School of Medicine, Louisville, KY, United States of America

* murthyboston@gmail.com, b.murthymedbio@gmail.com

**Data Availability Statement:** Authors declare that the data supporting the findings of this study are

## Abstract

Biallelic mutations in *ACP5*, encoding tartrate-resistant acid phosphatase (TRACP), have recently been identified to cause the inherited immuno-osseous disorder, spondyloenchondrodysplasia (SPENCD). This study was undertaken to characterize the eight reported missense mutations in *ACP5* associated with SPENCD on TRACP expression. *ACP5* mutant genes were synthesized, transfected into human embryonic kidney (HEK-293) cells and stably expressing cell lines were established. TRACP expression was assessed by cytochemical and immuno-cytochemical staining with a panel of monoclonal antibodies. Analysis of wild (WT) type and eight mutant stable cell lines indicated that all mutants lacked stainable enzyme activity. All *ACP5* mutant constructs were translated into intact proteins by HEK-293 cells. The mutant TRACP proteins displayed variable immune reactivity patterns, and all drastically reduced enzymatic activity, revealing that there is no gross inhibition of TRACP biosynthesis by the mutations. But they likely interfere with folding thereby impairing enzyme function. TRACP exists as two isoforms. TRACP 5a is a less active monomeric enzyme (35kD), with the intact loop peptide and TRACP 5b is proteolytically cleaved highly active enzyme encompassing two subunits (23 kD and 16 kD) held together by disulfide bonds. None of the mutant proteins were proteolytically processed into isoform 5b intracellularly, and only three mutants were secreted in significant amounts into the culture medium as intact isoform 5a-like proteins. Analysis of antibody reactivity patterns revealed that T89I and M264K mutant proteins retained some native conformation, whereas all others were in "denatured" or "unfolded" forms. Western blot analysis with intracellular and secreted TRACP proteins also revealed similar observations indicating that mutant T89I is amply secreted as inactive protein. All mutant proteins were attacked by Endo-H sensitive glycans and none could be activated by proteolytic cleavage *in vitro*. In conclusion, determining the structure-function relationship of the SPENCD mutations in TRACP will expand our

available within the paper and its Supporting Information files.

**Funding:** This work received partial support from the following sources: University Grants Commission - Special Assistance Programme (UGC - SAP) and Department of Science and Technology - Promotion of University Research and Scientific Excellence (DST- PURSE). The funders had no role in study design, data collection and analysis, decision to publish, or preparation of the manuscript.

**Competing interests:** The authors have declared that no competing interests exist.

understanding of basic mechanisms underlying immune responsiveness and its involvement in dysregulated bone metabolism.

## Introduction

Tartrate-resistant acid phosphatase (type-5 acid phosphatase; TRACP, EC 3.1.3.2.) is a ~35 kD metalloenzyme with a mixed valency di-iron center, required for catalytic activity [1]. TRACP is expressed primarily by differentiated cells of the mononuclear phagocyte system including osteoclasts, macrophages and dendritic cells (DC) [2]. It has a long history of clinical relevance as a cytochemical marker for hairy cell leukaemia, and as a serum marker for osteoclastic bone resorption and, more recently, chronic inflammation [3]. TRACP functions *in vitro* cleaving unnatural phosphates like p-nitro-phenyl phosphate but its biological role in bone resorption and immune responses may be different [1]. One natural substrate is osteopontin (1). TRACP exists as two isoforms, which are derived by differential post-translational processing of a central regulatory loop peptide [4]. TRACP5a, a monomeric protein, with the intact loop peptide, has a lower pH optimum of ~5 and specific activity of ~100 U/mg. TRACP5b is proteolytically cleaved into a 23 kD and 16 kD disulphide linked heterodimer with a pH optimum of ~6 and specific activity of ~1000 U/mg [5]. In addition, isoforms 5a and 5b are differentially compartmentalized. In macrophages and dendritic cells, only isoform 5a is secreted from cells; isoform 5b remains intracellularly [6]. Therefore, serum TRACP5a serves as a marker for systemic macrophage functions and chronic inflammatory activity [3]. In osteoclasts, however, isoform 5b is released into the circulation with other matrix products during bone resorption, thus serving as a marker for osteoclast activity [6]. The regulatory signals that govern this differential processing are not fully understood.

The complex inherited disease spondyloenchondrodysplasia (SPENCD) is a recently described disorder comprising of craniofacial, skeletal, neurological and autoimmune manifestations [7, 8, 9]. More specifically, these include skeletal dysplasia and radiolucent metaphysical lesions which arise from biallelic mutations of *ACP5* gene, which encodes TRACP enzyme [9–11]. We and others have shown earlier [12, 13, 14] that mutations of *ACP5* can cause autoimmune cytopenia, immuno-osseous dysplasia, spasticity with leukodystrophy, systemic lupus erythematosus (SLE), Moyamoya syndrome and Sjogren's syndrome [15–17]. Other salient features of SPENCD include retardation of growth with developmental delays, clumsy movements and specific neurological symptoms such as intracranial/cerebral calcifications. Increased expression of type-I INF regulated genes which tracks parallel with extra skeletal abnormalities has also been observed [12–14].

*ACP5* is transcribed from a single gene with 5 exons. Three alternate promoters exist within the first three exons (E1A, E1B and E1C) [18]. The TRACP protein is translated from exons 2 to 5. Molecular studies such as promoter analysis and assignment of chromosomal localizations have been under taken by Reddy et al., [19] for human and mouse genes. Molecular modelling of the eight-missense mutant TRACP proteins associated with SPENCD suggested that single amino acid substitutions could lead to protein destabilization [12–14]. In SPENCD, *ACP5* consists of partial or whole gene deletions and nonsense or missense single base substitutions. Seventeen distinct mutations have so far been reported by two groups including ours (4 deletions, 5 nonsense mutations and 8 missense single base changes) [9, 10]. Of the ten patients in whom TRACP activity or protein was studied, no detectable TRACP activity was lacking in cells (4 patients) or no circulating TRACP protein was found in serum (6 patients) (unpublished). The clinical presentation reinforces the concept that TRACP is a member of the growing number of molecules important in osteoimmunology, and may be a key

pathophysiological player. It may also be a therapeutic target in metabolic bone diseases [20], immune disorders [21, 22] and cancer [23, 24].

The overall purpose of the study is to clone and express all the missense *ACP5* genes in human embryonic kidney-293 (HEK-293) cells which are analogous to those observed with SPENCD related mutations in humans [12–14]. The resultant TRACP protein products were characterized in human cell lines to define the effects of the specific amino acid changes and provide direct evidence for the causal mechanism of TRACP deficiency in SPENCD patients. From a practical perspective, these clinically relevant mutations were exploited in human derived stable cell lines to learn more about the specific epitopes targeted by unique monoclonal antibodies to TRACP enzyme developed in our laboratory.

## Materials and methods

### Transformation of mutant genes and subcloning

Wild type and eight missense *ACP5* mutant genes [12, 13] were synthesized and cloned directly into a proprietary pJ602 vector containing the cytomegalovirus (CMV) promoter using ampicillin and Zeocin resistance genes for selection (ATUM formally DNA 2.0, Menlo Park, CA). *ACP5* genes contained single base changes leading to amino acid substitutions: K52T (c.155A>C), T89I (c.266C>T), G109R (c.325G>A), L201P (c.602T>C), G215R (c.643G>A), D241N (c.721G>A), N262H (c.784A>C), and M264K (c.791T>A) (**Fig 1**). Single colonies of transformed *E. coli* were isolated on LB plates containing 100 μg/ml of ampicillin (LB-amp). Plasmid DNAs were prepared from 500 ml of LB-amp broth cultures using Maxi-prep kits (Qiagen Inc, Valencia, CA). Human monocyte derived and recombinant TRACP proteins were prepared as described earlier [25, 26].

### Transient transfection and generation of stable cell lines

HEK-293 cells were transfected with one to two μg of plasmid DNAs using Fugene 6 reagent (Roche Applied Science, Indianapolis, IN) according to recommended procedures. Cells were screened for TRACP expression after 48 hours, after which cytochemical and immunocytochemical techniques [27, 28] were carried out as described below. Transiently transfected cells and their supernatant media were harvested after 72 hours. Cytocentrifuge smears were prepared from the cells, air dried and stored at room temperature until used for staining. Media were stored in aliquots at –70°C until used for analyses. The cell lysates were prepared at $10^7$ cells/ml in a buffer of 10 mM Tris (hydroxymethyl) aminomethane (TRIS), 1 mM Ethylene Diamine Tetraacetate (EDTA), 1 mM Ethylene Glycol-bis-(aminoethyl ether) Tetraacetate (EGTA), 1% Nonidet P-40 (NP-40), 300 mM NaCl and protease inhibitor cocktail (2 mM phenyl methyl sulfonyl fluoride (PMSF) / 10 μg/ml aprotinin / 10 μg/ml leupeptin, at pH 7.4). After 30 minutes on ice, the lysates were cleared of insoluble debris by centrifugation. Total protein in cell lysates was determined by a commercial Coomassie Blue dye binding method (Bio-Rad, Richmond, CA) using bovine serum albumin (BSA) as standard. For all immunoassays and immunoblots, the cellular protein input for mutant TRACP was normalized to that of wild type (WT) TRACP-expressing cells (4 μg/μl). Soluble proteins were stored at –70°C until used for analyses. Stable transfectants were selected using 10 to 100 μg Zeocin/ml. After 7–14 days, growing colonies were picked and expanded in the presence of Zeocin and re-picked until the growing clones were 100% positive for TRACP expression by immuno-cytochemical staining. Stable clones are grown in the continuous presence of 10–100 μg Zeocin/ml. Media and cell lysates were also prepared from the stable transfectant clones.

```
        MDMWTALLIL   QALLLPSLAD   GATPALRFVA   VGDWGGVPNA   PFHTAREMAN
                    Leader peptide
         51
        AKEIARTVQI   LGADFILSLG   DNFYFTGVQD   INDKRFQETF   EDVFSDRSLR
        T                                                I
        101               *                                       *
        KVPW YVLAGN   HDH LGNVSAQ   IAYSKISKRW   NFPSPFYRLH   FKIPQTNVSV
              R
        151
        AIFMLDTVTL   CGNSDDFLSQ   QPERPRDVKL   ARTQLSWLKK   QLAAAREDYV
                              Loop peptide
        201
        LVAGHYPVWS   IAEHGPTHCL   VKQLRPLLAT   YGVTAYLCG H   DHNLQY LQDE
        P                     R                              N
        251
        NGVGYVLSGA   GNFMDPSKRH   QRKVPNGYLR   FHYGTEDSLG   GFAYVEISSK
                              H  K
        301
        EMTVTYIEAS   GKSLFKTRLP   RRARP
```

**Fig 1. Amino acid sequence of human TRACP.** Amino acid substitutions due to the observed missense mutations are shown below the wild (WT) sequence. Functionally important regions, including the leader peptide and regulatory loop peptides are underlined. Iron-binding two domains are represented in bold italic. N-linked glycan acceptor asparagines are noted by an asterisk (*) on the top of the residue.

## Control dendritic cell preparations

Dendritic cells (DC) were differentiated from blood monocytes as described earlier [25]. Monocytes were isolated by density gradient centrifugation of buffy coat cells from a de-identified therapeutic phlebotomy specimen from a patient with hemochromatosis. Enriched monocytes were cultured in 10 cm dishes at $10^6$ cells/ ml in RPMI-1640 medium supplemented with antibiotics, 10% fetal bovine serum, 25 ng/ml GM-CSF and 20 ng/ml interleukin 4. An additional dose of cytokines was added at day four. On day six, non-adherent cells were decanted, and the dishes were rinsed gently once with Hank's balanced salt solution (HBSS). Five ml of HBSS containing 2 mM EDTA was added to detach the differentiated DC. Cells were washed 3 times in HBSS, pelleted and lysed at $10^7$ cells/ml and the protein content was determined as described above. TRACP secreted to medium was enriched by one-step purification by ion exchange chromatography [18, 19]. One hundred-twenty ml of DC culture medium after day six was collected and adjusted to pH 5.0, clarified by centrifugation and applied to a 5 ml column of SP Sepharose (GE Healthcare). After washing the column with washing buffer (10 mM Na acetate/50 mM NaCl), the bound TRACP was eluted with elution buffer 10 mM Na acetate/1.0 M NaCl, and ten 1.0 ml fractions were collected. TRACP activity was determined using 10 μl of each fraction in 200 μl of 10 mM, 4-Nitrophenyl phosphate in 100 mM Na acetate/50 mM sodium tartrate buffer, pH 5.5. The active peak fractions were pooled and dialyzed against 50 mM Na acetate buffer pH 5.5 containing 100 mM NaCl/2% glycerol.

## TRACP cytochemistry and immunocytochemistry

Cytocentrifuge smears were prepared and stained for TRACP activity using naphthol ASBI-phosphate as substrate and hexazotized pararosaniline as coupler at pH 5.5 according to

published methods [27]. TRACP proteins were stained immuno-cytochemically using mono-clonal antibody, Mab220, for intact TRACP5a like isoform, Mab9C5 for total TRACP protein and 5b isoform as well [28]. Control smears were also stained using IgG from non-immune ascites. Fixed smears were subjected to heat-induced epitope retrieval for 30 minutes at 70°C in a commercial solution of EDTA at pH 8.0 (Invitrogen Corp, Carlsbad, CA), followed by 15 seconds permeabilization in 0.2% Triton X-100/PBS prior to immunochemical staining.

**Immunoassay for TRACP activity and protein.**   Antibodies used for immunoassays were all developed in our laboratory [25–28]. Monoclonal antibodies, 14G6 (Mab14G6) and Mab162 target independent conformational epitopes on both isoforms 5a and 5b and are used to measure TRACP5b activity and total TRACP protein in native conformation. These epitopes are destroyed by heating in sodium dodecyl sulfate (SDS) and 2-mercaptoethanol. Mab220 targets the trypsin-sensitive regulatory loop peptide present only in isoform 5a and reacts with both native and denatured forms of the protein. Mab220 is used as capture antibody to measure isoform 5a activity and protein specifically. Mab9C5 targets a linear epitope in the *C*-terminal part of both isoforms 5a and 5b after heat denaturation. It is suitable for Western blot analysis of both TRACP5a (35 kD) and 5b (16 kD) but does not react with native enzyme. Mab9C5 is used as capture antibody to measure denatured or non-native precursor TRACP protein [25, 26].

TRACP isoform activities were measured in growth media, and cell lysates according to our published methods [25, 26]. After immobilization of TRACP with Mab220 (isoform 5a) or Mab14G6 (total TRACP), the bound enzyme activity was determined using 4-nitrophenyl phosphate at pH 5.8 and 6.1 respectively. Assays were standardized against a serial dilution of 4-nitrophenolate equivalent to 0.16 to 10 IU (μmol 4-Nitrophenyl phosphate (4-NPP) hydrolysed/minute/liter). TRACP isoform proteins were immobilized with Mab220 (native isoform 5a), Mab14G6 (total native TRACP) or Mab9C5 (total denatured TRACP including 5b isoform). Immobilized native TRACP proteins were detected with a unique anti-TRACP Mab162 conjugated to horseradish peroxidase (Mab162-HRP). Immobilized denatured TRACP protein was detected with Mab220-HRP. Peroxidase activity was determined with o-Phenylene diamine dihydrochloride and $H_2O_2$ [25, 26]. Native TRACP protein assays were estimated using serial dilutions of partially pure serum TRACP5a (0.08 to 5 ng/ml). Results for the denatured TRACP assay are expressed as $A_{490}$ absorbance values.

**Western blotting.**   To determine the effects of TRACP mutations the expressed proteins from culture medium (10 μl) and cell lysates ($10^7$ cells/ml) were subjected to Western (immuno) blot analysis using Mab220 (TRACP5a) and Mab9C5 (TRACP 5a and 5b) as probes [28]. Ten μl medium was used for analysis of all cell culture medium samples. Protein amounts in lysates were normalized to that in 2 μl of WT. Samples were heated to 100°C for 2 minutes under reducing conditions using 2-mercapto ethanol, and electrophoresed in 12% SDS-polyacrylamide gel electrophoresis (PAGE) gels. After electrophoresis proteins were transferred onto polyvinylidene fluoride (PVDF) membranes for immuno-detection [29]. The membranes were blocked with 3% milk powder for 1 h at room temperature in 1% Tween 20 in TBS (20 mM Tris, pH 7.5, 0.5 M NaCl) and washed three times in TBST (TBS + 0.05% Tween 20). Primary antibodies were used in TBST as well and washed again for three times followed by secondary antibody conjugated with alkaline phosphatase (rabbit anti-mouse IgG). Immuno blot color development was carried out with Nitro Blue Tetrazolium chloride/ 5-bromo-4-chloro-indol-3-yl phosphate *p*-toluidine salt following the modified method of Brenan and Bath [30] based upon original methods of Van Noorden [31] and McGadey [32]. Blots finally washed with water, fixed in EDTA and dried at room temperature.

**Deglycosylation and proteolysis of TRACP proteins.**   To determine if mutations had cause altered post-translational modifications, cell lysates and supernatant media were exposed

to endoglycosidase-H (Endo-H; Boehringer Mannheim Biochemicals, Indianapolis, IN) or peptide N-glycosidase F (PNGaseF; New England Biolabs, Beverly, MA) after mild denaturation. Amounts normalized to protein in 2 μl WT cell lysate (4 μg/μl) or 10 μl culture medium were mixed in triplicate with 1 μl PNGase F denaturing buffer (NEB) and water was added to make 10 μl. Samples were boiled for 10 minutes to denature protein and expose oligosaccharides. Then 0.5 mU (1 μl) Endo-H and 1 μl 10X buffer (500 mM Na citrate / 10% NP-40, pH 5.5), or 500U (1 μl) PNGase F and 1 μl 10 X buffer (500 mM NaPO$_4$ / 10% NP-40, pH 7.0), or 1 μl 10X Endo H buffer were added to the denatured samples. Digestion was carried out at 37˚C for 60 minutes. An equal volume of SDS-PAGE reducing sample buffer was added and the samples were stored at −70˚C until used for Western blot analysis with Mab9C5.

### Limited proteolysis of TRACP proteins with trypsin

To examine if mutant intracellular TRACPs could be activated by proteolysis *in vitro* from isoform 5a-like to 5b-like enzymes, TRACP-HEK lysates in 2 μl WT cells or 10 μl culture medium were digested with 200 mU solid-phase trypsin-agarose (Sigma Chem Co, St. Louis, MO) or control BSA-conjugated agarose (2 mg BSA /ml packed agarose) made with Amino Link Plus coupling gel (Pierce Chemical Co., Rockford, IL) for 2h at room temperature. Reactions were made to 50 μl with 10 mM Tris / 30 mM NaCl / 0.1% Triton X-100, pH 7.0. After incubation, the solid-phase trypsin and BSA were removed by centrifugation. Digests were subjected to Western blot analysis with Mab9C5.

### Statistical analysis

All data were analysed using GraphPad Prism Version 8.0.1. For all measurements, student T-test was used to assess the statistical significance between groups. All experiments were carried out at least three times before arriving at statistical differences. Statistically significant difference was as follows: **** highly significant ($p < 0.00005$), *** moderately significant ($p < 0.0005$), ** significant ($p < 0.005$), * less significant ($p < 0.05$).

## Results

### Amino acid sequences of TRACP holo enzyme and its structural features

**Fig 1** illustrates the entire amino acid sequence of human holo-TRACP with leader and loop peptides. Missense mutations observed in SPENCD are depicted below the wild (WT) sequences. In the homozygous or heterozygous state, these mutations result in TRACP deficiency in cells and serum resulting in a complex heritable disease SPENCD [12–14]. Apart from the leader peptide (1–21 residues), when the loop peptide is removed it increases the enzyme activity converting TRACP 5a into 5b isoform. In **Fig 1** the loop peptide resides between 162–182 residues. The iron binding domains of TRACP mediating ROS generation are indicated by bold italics in **Fig 1** which lie between residues 105–113 and 240–247. Two distinct *N*-glycosidic linkage acceptor asparagines (116 and 147) exist in TRACP as indicated. This type I linkage with the canonical amino acid triplets NVS (**Fig 1**; amino acids 116–118 and147-149) are well conserved and destined to classical ER mediated processing of linked carbohydrate moieties.

### Recombinant mutant TRACP expression

Transient (72 hr) and stable expression of mutant TRACP proteins were confirmed by cytochemical staining for TRACP activity. Immuno-cytochemical stainings were carried out using Mab220 (TRACP5a) and Mab9C5 (total TRACP). As shown in **Fig 2** WT TRACP only

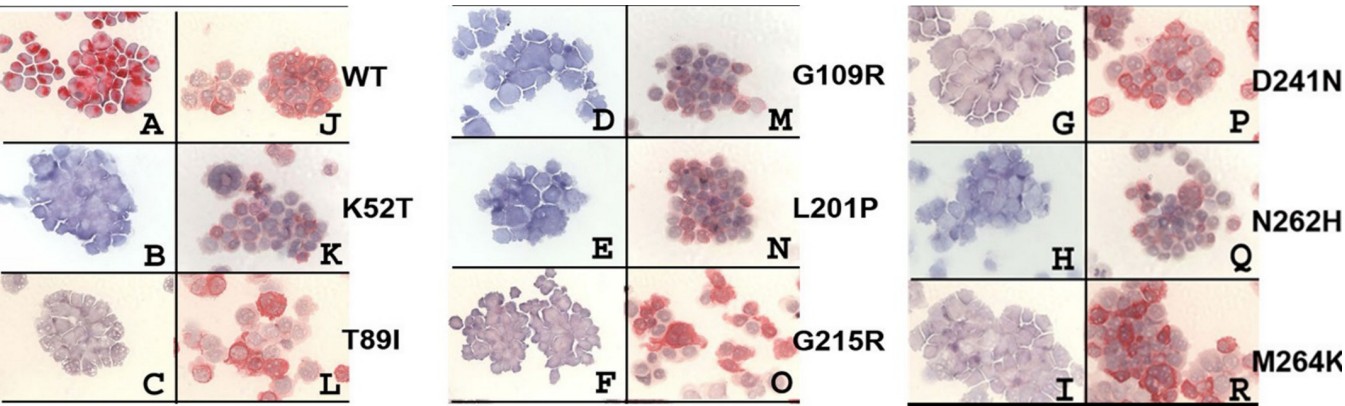

**Fig 2. Cytocentrifuge smears were prepared from WT and mutant TRACP cells at 72 hours after transfection.** Smears were fixed and stained directly for TRACP enzyme activity (A to I) or immuno-cytochemical staining for intact TRACP protein with Mab220 (J to R) was carried out as a probe. Only wild (WT) TRACP was active, all mutants cells expressed as inactive proteins. Experiments were repeated with five slides at a time and the best stained pictures were presented.

expresses strong activity while all mutants lacked stainable activity. However, immuno-cytochemical staining for intact TRACP protein (Mab220) showed that expression was similar in all transient transfectants and that stain intensity was strong for all mutants almost equivalent to that of WT TRACP. Immuno-staining with Mab9C5 demonstrated similar results analogous to those with Mab220 (not shown). These results indicate no gross inhibition of TRACP polypeptide biosynthesis by the mutations; but the mutations likely interfere with enzymatic function(s).

## Expression of intracellular and secreted recombinant TRACP proteins

In contrast to the immuno-cytochemical results, quantitative immunoassay for TRACP isoforms secreted into medium (**Fig 3**) and expressed intracellularly (**Figs 4A and 4B** for TRACP activity 5a and 5b respectively; **Fig 4C and 4D** for total protein) initially showed an almost complete lack of TRACP expression in six out of eight missense mutations examined. Only T89I and M264K mutants were detectable to an appreciable degree as inactive proteins (**Fig 4C and 4D**). This is explained by the fact that Mab14G6 used to capture total TRACP and Mab162-HRP used as detection antibody in immunoassays react with only the native conformation of TRACP [33]. Therefore, the T89I and M264K mutations likely result in the synthesis of enzymes with at least partial native conformations conserved at the Mab14G6 and Mab162 epitopes, in addition to the Mab220 loop peptide epitope specific to TRACP5a. An immunoassay was devised to identify the so-called "denatured" TRACP using Mab9C5 to capture TRACPs and Mab220-HRP was used to detect 5a form. This assay revealed that all mutant proteins were detectable intracellularly to a similar level (**Fig 5**, right, y axis = cells), and that K52T and T89I TRACP were uniquely secreted into the culture medium (**Fig 5**, left, y axis = medium) as inactive "denatured" TRACP proteins (**Fig 5**). Since WT TRACP is also reactive with Mab9C5, this assay also detects normal immature precursor protein. Our results confirm that all eight TRACP missense mutations result in inactive proteins based on immunochemical assays. It suggests that the destabilization of tertiary structures and misfolding do not allow synthesis of an active enzyme with appropriate protein integrity. Only T89I and M264K were synthesized with intact native epitopes recognized by Mab14G6 and Mab162 and secreted into medium. However, M264K was not detected in the media by immunoassay using

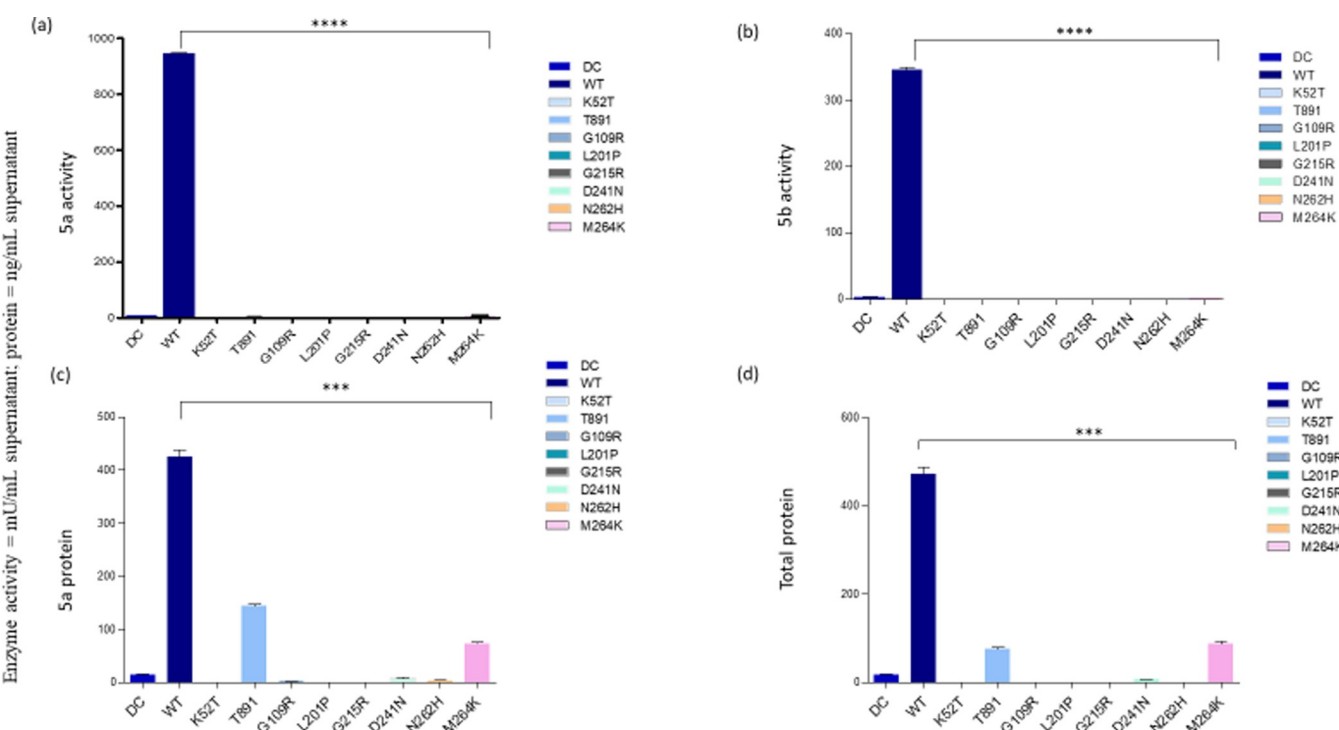

**Fig 3. Analysis of recombinant mutant TRACP proteins secreted to medium.** Figs (a), (b), (c), and (d) represent the secreted 5a, 5b protein and total protein levels observed (Different mutations tested are indicated on the X-axis). Multiple-group comparisons were performed using one-way analysis of variance (ANOVA) for all samples. The data is presented as mean ± S.E.M. ****, P <0.00001; ***, P <0.0001; Experiments were repeated three times. The Y axis represents the activity observed with different mutant cells in various conditions along with Dendritic cells (DC) and WT as controls.

Mab9C5. Conversely, K52T was secreted to medium, perhaps as an immature precursor reactive with Mab9C5, but was not reactive to either Mab14G6 or Mab162.

## Mutant TRACP protein processing

Studies on post translational processing of TRACP mutations carried out by Western blot analysis indicated that WT TRACP exists intracellularly as a mixture of intact 5a-like protein and cleaved 5b protein (**Fig 6**, Lanes 2–3) whereas all mutant TRACPs existed solely as uncleaved proteins both in media and cell lysates (**Fig 6**, Lanes 4–19). In the top panel (**Fig 6**) the blot was probed with Mab220 and in the lower blot was probed with Mab9C5. Western blot analysis revealed that the level of secreted TRACP in media was dramatically reduced in all mutations except T89I (**Fig 6**; Lane 6 both top and bottom figures). Analysis of mutant K52T by immunoassay and Western blot analysis of secreted TRACP with Mab9C5 did not show any correlation: i.e., it showed a strong signal in immunoassay (**Fig 5**) and a weak signal in blots (**Fig 6**; Lane 4, sample marked as "m" underneath the figure).

## Deglycosylation of TRACP proteins with Endo H and PNGase F enzymes

We next investigated whether the amino acid substitutions in TRACP mutations, which are not part of the *N*-glycan acceptor sites (21–25), affected the glycosylation of mutant human TRACPs. The WT and mutant proteins that are secreted into medium have similar glycosylation patterns, similar to the natural TRACP secreted by dendritic cells (DC) (**Fig 7**, Lanes 2–4 for DC cells; Lanes 5–7 for WT cells) with both Endo-H sensitive and resistant glycans (**Fig 7**; upper figure for TRACP from medium; lower figure for TRACP proteins derived from cells.

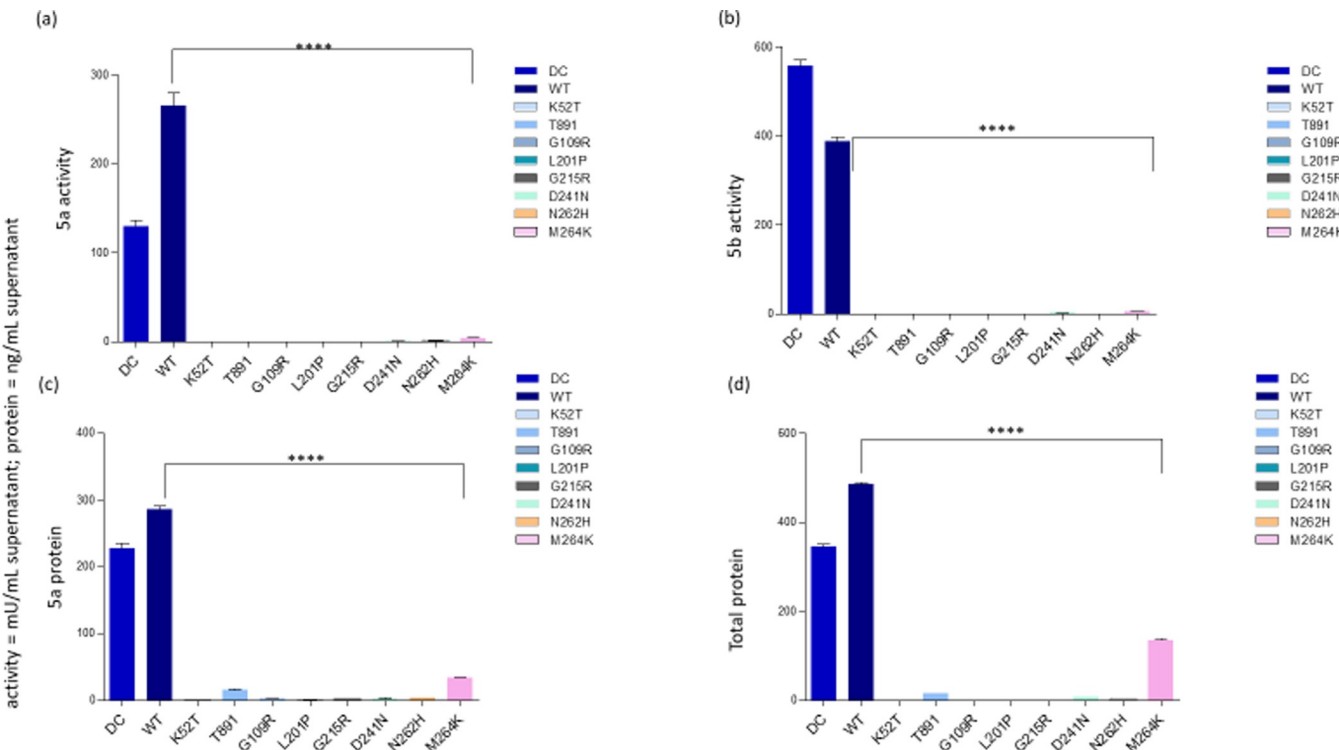

**Fig 4. Intracellular expression of recombinant mutant TRACP.** Figs (a) and (b) represent the intracellular 5a, 5b enzyme activity and Figures c and d represent total 5a and 5b protein levels. Figures are depicted with DC, WT as controls along with different mutant cell activities (mutation amino acids are indicated below the X-axis). Details are similar to those indicated in the legend to Fig 3.

The recombinant TRACP remaining intracellularly bears more Endo H sensitive glycans compared to natural TRACP, particularly those with mutations (**Fig 7**; Lanes 8–31). Mutant K52T has a higher molecular weight compared to all other TRACP proteins (**Fig 7**; Lane 8) prior to deglycosylation. However, after removal of oligosaccharides, the polypeptide was the same size as other TRACP proteins (**Fig 7**, Lanes 9–10).

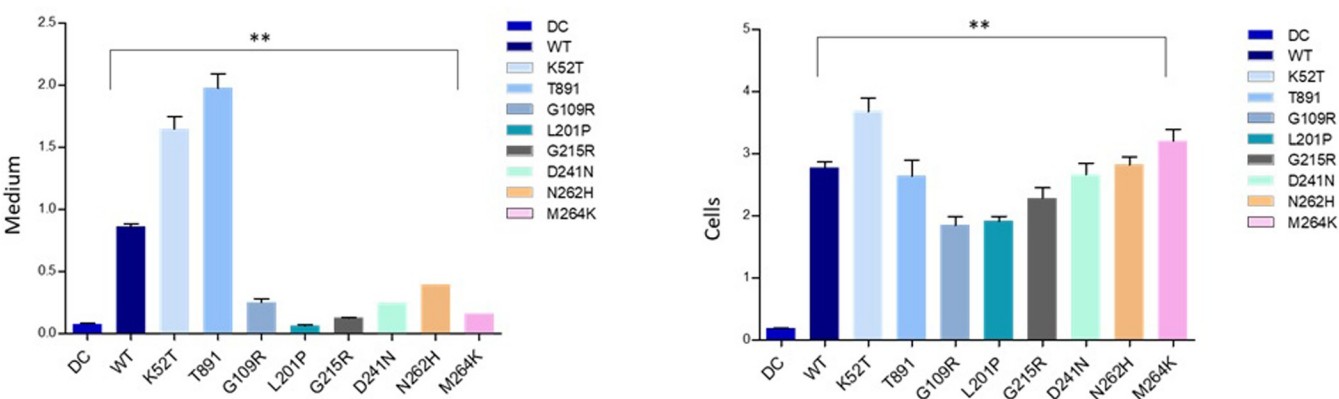

**Fig 5. "Denatured" or precursor mutant TRACP protein expressed by HEK-293 cells.** Denatured TRACP was measured by immunoassay using Mab 9C5 to capture TRACP and Mab 220-HRP was used as secondary antibody to detect bound proteins. Y axis = Medium = Medium supernatants (Left figure); Cells = cell lysates; right figure (protein in samples were normalized to WT type samples; in three separate experiments). Figures represent total protein levels observed with different mutations from medium and cells (mutation amino acids are indicated on X-axis). Details are similar to those indicated in the legend to Fig 3.

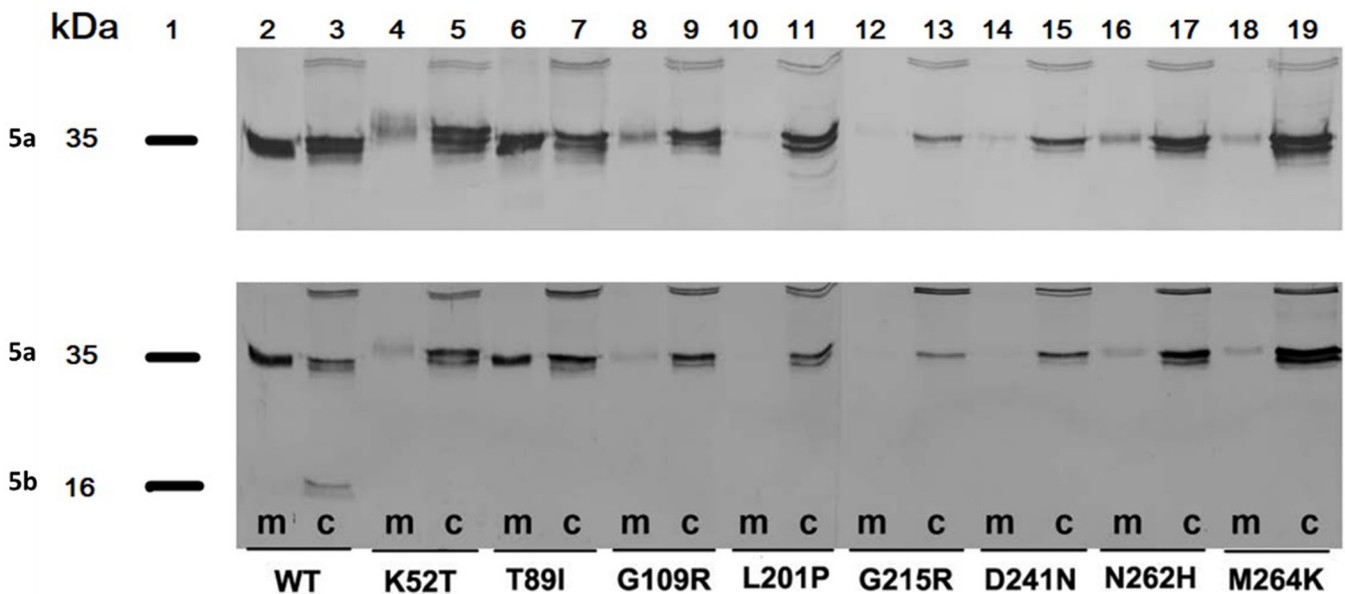

**Fig 6. Western blot analysis of mutant TRACP proteins secreted to culture media (m) and expressed intracellularly (c).** All mutants express intact 35 kD 5a-like proteins intracellularly; only WT TRACP is present as a processed protein (16 kD fragment shown, Lane 3- WT,c). All mutants are secreted at low levels except T89I, which is secreted in similar amounts to WT. Lanes 8 and 16 in the upper panel, mutants G109R and N262 show mild secretary forms as recognized by Mab220. Duplicate blots were probed with Mab220 (isoform 5a-specific antibody, upper panel) or with Mab 9C5 (5b specific antibody, bottom panel) to detect both intact (5a) and cleaved (5b) TRACP isoforms. Lane 1 indicates the molecular weights of TRACP 5a and 5b. Lanes 2 to 19 indicate samples derived from medium (m) and cells (c).

## Limited proteolysis of TRACP proteins by trypsin

In order to study the conversion of proteolytically cleaved inactive mutant TRACP proteins into active TRACP 5b like enzymes, we used trypsin to fragment them along with wild (WT) type TRACP (**Fig 8**; Lanes 2–5). Trypsin successfully digested all intracellular mutant TRACP monomeric proteins (**Fig 8**; Lanes 6–21). The 16 kD fragment of isoform 5b was clearly evident in trypsin-treated WT samples—Lanes 2 and 3 for TRACP from medium and lanes 4 and

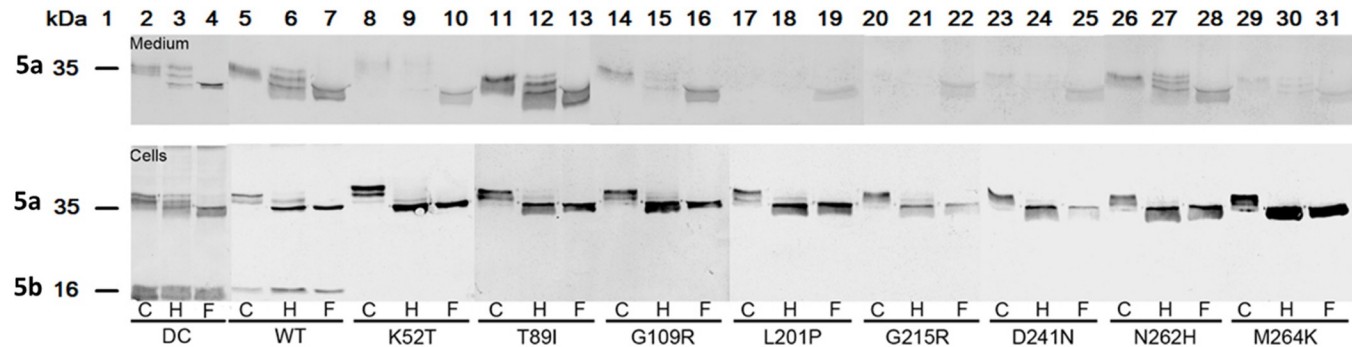

**Fig 7. Western blot analysis of natural dendritic cell (DC), recombinant WT and mutant samples.** Both DC and WT were used as controls. Mutant TRACP proteins, undigested (C) or after deglycosylation with Endo H enzyme (H, high-mannose and hybrid oligosaccharides) or PNGase F enzyme (F, all N-linked oligosaccharides) were run along with other mutant samples. In the upper panel all mutants faintly demonstrate some bands excepting T89I which showed strong oligomeric bands after deglycosylation. Blots were probed only with Mab9C5 to detect both intact and cleaved forms. Lane 1 indicates the molecular weights of TRACP 5a and 5b (35 kD and 16 kD respectively). Lanes 2 to 31 indicate various samples treated with Endo H and PNGase F (Endo H = H; PNGase F = F).

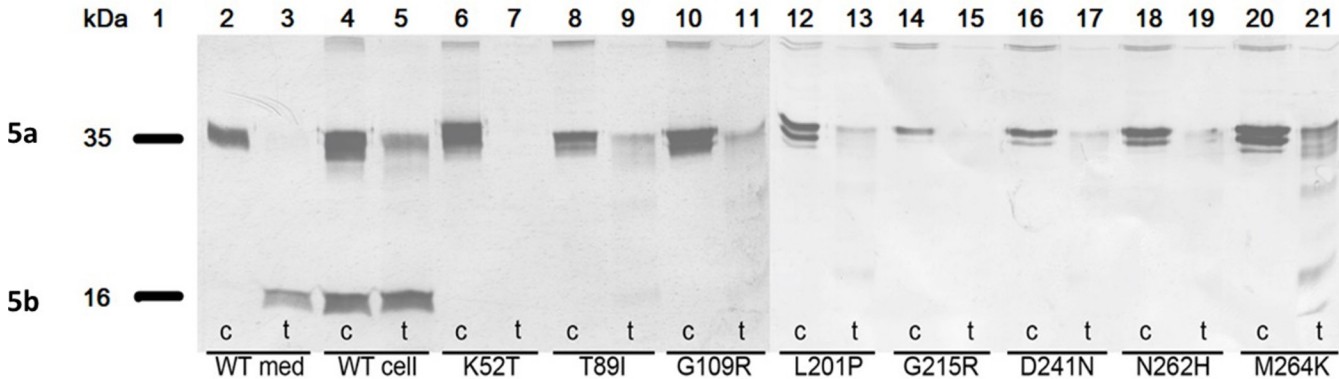

**Fig 8. Western blot analysis of WT and mutant TRACP proteins in cell lysates after trypsin treatments.** Undigested control (c) and digested/treated (t) samples were exposed with trypsin (Lanes 2–21). Normal WT samples only show 16 kD forms from both medium (Lanes 2, 3) and cells (Lanes 4, 5) as recognized by Mab9C5. Lane 1 illustrates the molecular weights of TRACP 5a and 5b (35kD and 16kD). Lanes 6 to 21 indicate various mutant samples treated with trypsin. These lanes demonstrate that most of these mutants are incapable of generating 16 kD isoforms unlike WT (Lanes 2–5) samples. Only M264K shows faint multiples bands after fragmentation by trypsin as recognized by Mab9C5 (Lane 21).

5 for TRACP from cells (intracellular). Although 16 kD TRACP is vaguely visible in T89I, G109R, L201P and M264K mutants (**Fig 8**; Lanes 8–13 and 20–21), additional larger proteolytic fragments were also vaguely visible in the M264K mutant (Lanes 21). As expected, proteolysis did not convert TRACP5a to strong TRACP5b bands in blots. There was also no increase in the enzyme activity or pH optima of the TRACP mutants as observed in **Fig 9** both at pH 5.2 and 6.1 except for controls samples, DC and WT. Both top and bottom figures (**Fig 9A and 9C**) on the left denote untreated controls whereas those on the right (**Fig 9B and 9D**) represent trypsin treated samples. These results suggest that the missense mutant proteins (which are probably misfolding) may be more susceptible to trypsin degradation *in vitro*. Complete degradation of TRACP proteins are clearly visible in lanes 7, 9 11, 13, 15, 17, and 19 even with limited proteolysis with trypsin.

## Discussion

Tartrate-resistant acid phosphatase plays an important pathophysiologic role in osteoimmunology as evidenced by mice bearing loss-of-function mutations in *ACP5* [20, 21]). As the single known cause for SPENCD, a specific human disease with a bone and immune phenotype, loss-of-function TRACP mutations represent a proof-of-principle experiment of nature [12, 13]. Patients with homozygous or compound heterozygous mutations display a variable degree of skeletal dysplasia, variable levels of neurological deficit, one or more autoimmune phenotypes and bone tumors [7, 8, 12, 13, 39]. Recently, it was discovered that TRACP may also have pathological consequences when over-expressed [24]. Functional genomics have implicated *ACP5* as one among six genes with verified oncogenic and pro-invasive capability in human malignant melanoma [24]. These clinical consequences of *ACP5* as a result of under and over expression imply that *ACP5* is tightly regulated. Computer modelling predicts that the *ACP5* missense mutations in SPENCD could lead to a destabilized protein structure, but direct analysis of mutant TRACP proteins have not been done so far. Hence, the results of these studies with eight mutant TRACP proteins are reported in this paper. The present study examines the effects of single amino acid substitutions related to SPENCD on protein biosynthesis *in vivo* and its enzyme activity (both intracellular and secreted isoforms). It also sheds light on the specificity of the anti-TRACP monoclonal antibodies used in these studies and identifies some

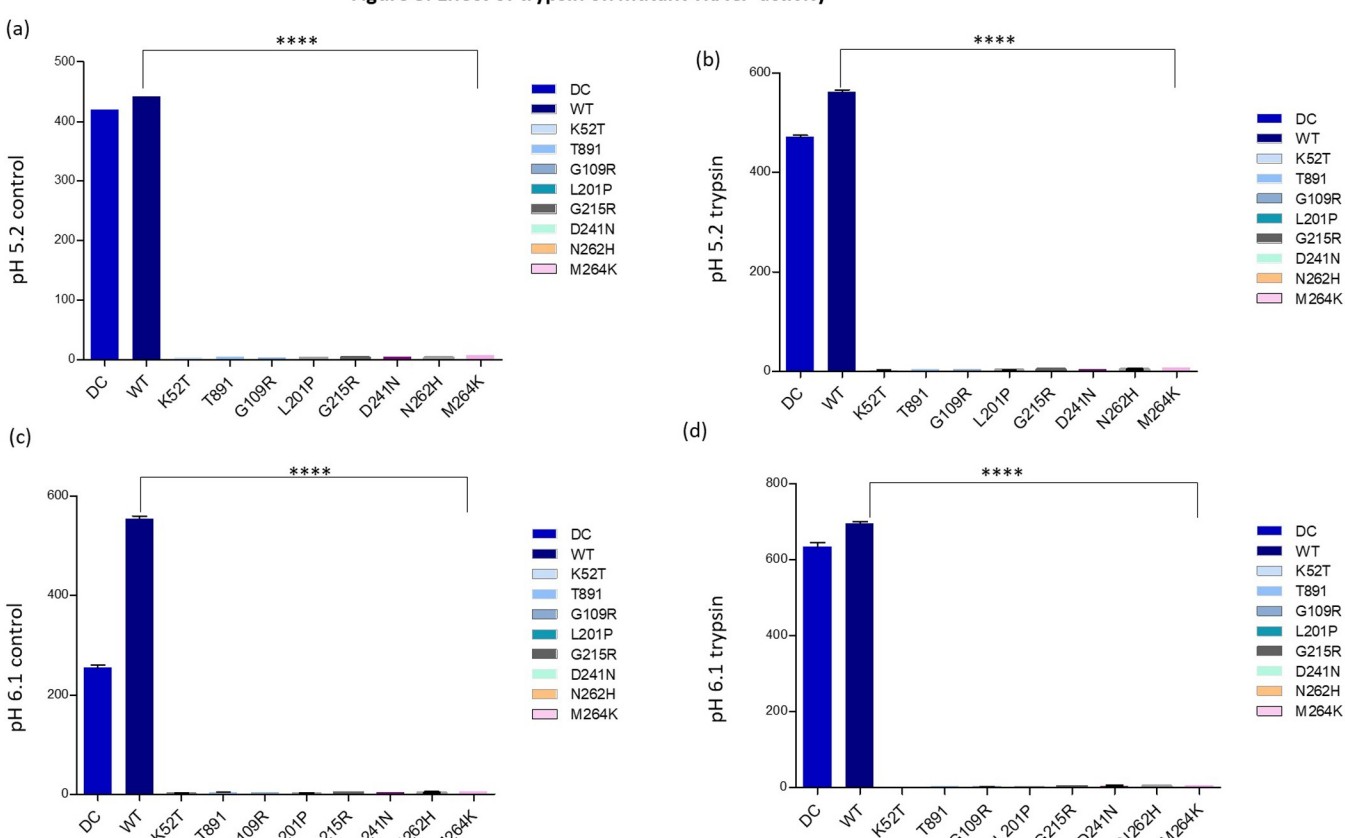

Fig 9. **Effect of trypsin treatments on mutant TRACP activities.** Enzyme samples from WT and DC media were used before and after trypsin treatments to estimate TRACP activities in two different pHs as described in Refs. 25 and 26. Controlled trypsin digestion increased the pH optimum from pH 5.2 to 6.1 and increased enzyme activities only in DC and WT samples. Low activities of mutant TRACPs were not affected by trypsin digestions. Experiments were repeated three times. Figs **a** (pH 5.2), **c** (pH 6.1) are untreated and served as controls. Whereas Figs **b** (pH 5.1), **d** (pH 6.1) represent enzyme activities observed after trypsin treatments. Details are similar to those indicated in the legend to Fig 3.

potential structural targets for inactivation of TRACP in instances where *ACP5* is pathologically over-expressed during oncogenesis.

TRACP5b remains in the intracellular endosomal compartment in most natural sources and in recombinant TRACP expressing cells, except osteoclasts [25, 26]. Osteoclasts uniquely release TRACP5b with other bone matrix products by transcytosis of endosomes during bone resorption [6]. The individual amino acid substitutions in mutant TRACP associated with SPENCD exhibited similar effects on activity and gross structure, but also had some variable effects on protein conformation as revealed by the pattern of immuno-reactivity with different antibodies (this study). First, all missense mutations caused inhibition of enzymatic activity either due to misfolding or modified structures. Second, immunoassays utilizing antibodies reactive to native conformational determinants showed that most mutants are undetectable in both cells and medium; with T89I and M264K being the exceptions. However, immuno-histo-chemical staining, Western blot analysis and immunoassays utilizing antibodies reactive with linear, denatured determinants/epitopes revealed that all mutant TRACPs were expressed abundantly intracellularly but were not processed by HEK-293 cells into highly active, hetero-dimeric secretory isoform 5b-like TRACP.

Immunoassays for native TRACP5a revealed that only T89I and M264K are secreted into medium as inactive proteins, whereas immunoassays for denatured TRACP5a showed

abundant secretion of K52T and T89I, but not M264K. Immunoblots of secreted TRACP with Mab220 and Mab9C5 both showed strong signals with T89I only. All other mutants were present in media in trace amounts. The anomalies causing method-based differential detection of the secreted forms of K52T and M264K remain unexplained. Since there is ample production of some of the mutants without enzyme activity, misfolding of proteins must therefore account for their inability to exert enzymatic functions. In all, five out of eight ACP5 mutations associated with SPENCD disease drastically interfere with TRACP exiting the cell. This explains the inability to detect TRACP protein in the serum of SPENCD patients [12, 13], or TRACP activity in their dendritic cells [13].

The site and type of glycosylation on TRACP is variable depending on the species and source of enzyme [34–36], and they have been assigned different functional roles. Human serum TRACP5a was originally defined by the presence of sialic acid [35]. Sialylation generally provides stability for a glycoprotein enzyme (TRACP) destined for secretion and extracellular function(s) [37]. Using a recombinant rat TRACP expression system, Wang *et al*. [35] demonstrated that at least one oligosaccharide is necessary for TRACP protein stability. In the present study all intracellular mutant TRACPs were *N*-glycosylated primarily with Endo H sensitive glycans (**Fig 7**, Lanes 8–31) whereas WT and DC TRACPs bore Endo-H resistant glycans (**Fig 7**, Lanes 2–4 especially for DC cells). Secreted TRACPs from all sources were similarly glycosylated in intact isoform-5a like proteins. The reduced amount of oligosaccharide processing and secretion of mutant TRACPs would be consistent with their destabilized structures. Failure to efficiently route to the Golgi for processing to add complex oligosaccharides will terminate eventual secretion.

Proteolytic processing of the loop peptide of monomeric TRACP is a critical mechanism for regulating enzymatic activity *in vivo* and *in vitro* [4, 5]. Limited proteolytic digestion with trypsin (**Fig 9**) was ineffective in converting the inactive mutant TRACPs to active enzymes. Proteolysis of the mutant proteins (**Fig 9**) may be responsible to generate inactive TRACP forms as measured for enzyme activity both at their optimal pH, 5.2 and 6.1. In all, the *ACP5* missense mutations associated with SPENCD generally lead to improper conformation, inefficient glycosylation, interrupted intracellular trafficking, and limited secretion. This finding explains the lack of detectable TRACP activity and protein in SPENCD patient sera and cells. Few mutant TRACP proteins that are secreted *in vitro*, for example T89I, probably do not survive mechanisms for removal of abnormal glycoproteins *in vivo*. It is uncertain why these mutant proteins with presumed abnormal structure escape intracellular degradation. Perhaps, we are detecting only a fraction of protein in transit during a continuous flux of precursor mutant TRACPs from the endoplasmic reticulum to the proteasome, without normal post-translational processing and endosomal compartmentalization of active isoform 5b or secretion of less active isoform 5a. Our findings contribute directly to understanding more completely the structure-function relationships in TRACP biochemistry. Further work on structural analysis of each TRACP mutant will reveal its precise functional relevance.

TRACP is emerging as a key pathophysiological component in human diseases, both when decreased or increased [24, 38]. Control of TRACP levels, its activity and compartmentalization are required for homeostasis in skeletal metabolism and immune responsiveness, and perhaps oncogenesis [24, 39, 40]. Therefore, research into the role of TRACP in immune regulatory circuits and the direct pathological effects of TRACP deficiency or over-expression should be rewarding. More detailed molecular analysis of mutant TRACP proteins could lead to design of specific drugs targeting TRACP. Currently in the pipe-line are drugs such as PSTP-3, 5-Me and Tiliroside, which target TRACP for osteoporosis [41, 42]. Also, appreciation of the direct effects of *ACP5* gene expression on diseases and pathways involving TRACP could lead to compounds that augment or mitigate these pathways upstream of TRACP [43].

## Supporting information

**S1 Fig.**
(PDF)

## Acknowledgments

We gratefully acknowledge Ms. Barbara Janckila for technical assistance with figures. This paper is dedicated to the memory of Prof. Dr. Lung T. Yam, MD (1936–2013), a pioneer in diagnostic biomarkers and discoverer of TRACP as a cytochemical marker for hairly cell leukemia. We truly appreciate his suggestions in the early phase of this work.

## Author Contributions

**Conceptualization:** Janani Ramesh, Latha K. Parthasarathy, Ranga N. Parthasarathy, Bhuvarahamurthy Venugopal.

**Data curation:** Janani Ramesh, Latha K. Parthasarathy, Ranga N. Parthasarathy, Bhuvarahamurthy Venugopal.

**Formal analysis:** Janani Ramesh, Ranga N. Parthasarathy, Bhuvarahamurthy Venugopal.

**Funding acquisition:** Bhuvarahamurthy Venugopal.

**Investigation:** Janani Ramesh, Ranga N. Parthasarathy, Bhuvarahamurthy Venugopal.

**Methodology:** Janani Ramesh, Latha K. Parthasarathy, Anthony J. Janckila, Ranga N. Parthasarathy, Bhuvarahamurthy Venugopal.

**Project administration:** Ranga N. Parthasarathy, Bhuvarahamurthy Venugopal.

**Resources:** Ranga N. Parthasarathy, Bhuvarahamurthy Venugopal.

**Software:** Janani Ramesh, Ranga N. Parthasarathy, Bhuvarahamurthy Venugopal.

**Supervision:** Bhuvarahamurthy Venugopal.

**Validation:** Janani Ramesh, Latha K. Parthasarathy, Ranga N. Parthasarathy, Bhuvarahamurthy Venugopal.

**Visualization:** Ranga N. Parthasarathy, Bhuvarahamurthy Venugopal.

**Writing – original draft:** Janani Ramesh, Ranga N. Parthasarathy, Bhuvarahamurthy Venugopal.

**Writing – review & editing:** Janani Ramesh, Farhana Begum, Ramya Murugan, Balakumar P. S. S. Murthy, Rif S. El-Mallakh, Ranga N. Parthasarathy, Bhuvarahamurthy Venugopal.

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
