## [Decision Letter · Decision Letter 0]

18 Aug 2019

PONE-D-19-20746

Characterisation of Acp5 Missense Mutations Encoding Tartrate-Resistant Acid Phosphatase Associated with Spondyloenchondrodysplasia

PLOS ONE

Dear Dr. Venugopal,

Thank you for submitting your manuscript to PLOS ONE. After careful consideration, we feel that it has merit but does not fully meet PLOS ONE’s publication criteria as it currently stands. Therefore, we invite you to submit a revised version of the manuscript that addresses the points raised during the review process.

We would appreciate receiving your revised manuscript by Oct 02 2019 11:59PM. To enhance the reproducibility of your results, we recommend that if applicable you deposit your laboratory protocols in protocols.io, where a protocol can be assigned its own identifier (DOI) such that it can be cited independently in the future. For instructions see: http://journals.plos.org/plosone/s/submission-guidelines#loc-laboratory-protocols

We look forward to receiving your revised manuscript.

Kind regards,

Dr. Sakamuri V. Reddy

Academic Editor

PLOS ONE

Journal Requirements:

1. Thank you for including your financial disclosure statement; "NO - The funders had no role in study design, data collection and analysis, decision to publish, or preparation of the manuscript."

Please provide an amended Funding Statement that declares *all* the funding or sources of support received during this specific study (whether external or internal to your organization) as detailed online in our guide for authors at http://journals.plos.org/plosone/s/submit-now.  

Please state what role the funders took in the study.  If any authors received a salary from any of your funders, please state which authors and which funder. If the funders had no role, please state: "The funders had no role in study design, data collection and analysis, decision to publish, or preparation of the manuscript."

Reviewers' comments:

Reviewer's Responses to Questions

**Comments to the Author**

1. Is the manuscript technically sound, and do the data support the conclusions?

Reviewer #1: Yes

Reviewer #2: Yes

2. Has the statistical analysis been performed appropriately and rigorously? 

Reviewer #1: Yes

Reviewer #2: Yes

3. Have the authors made all data underlying the findings in their manuscript fully available?

Reviewer #1: Yes

Reviewer #2: Yes

4. Is the manuscript presented in an intelligible fashion and written in standard English?

Reviewer #1: Yes

Reviewer #2: Yes

5. Review Comments to the Author

Reviewer #1: Major Comments:

1) In Results Section the Figures are not cited sequentially. It is cited like Fig. 2, 3 then 6A, 6B, Fig 3, then Fig. 7, Fig.4, Fig 5 and Fig. 8. Authors need to completely revise the Results section and explain the results according to the sequence of the figures or change the Figure numbers.

2) Figure legends are not written properly and completely. For example, in many Figure legends the results are explained and then experimental detail. There is no need to write the results in Figure legends. Please write the experimental details and number of repeat experiments.

3) Figures 6a and 6b should be numbered as Fig. 6 and Fig. 7 respectively.

3) It is not clear that why all mutants TRACPs were detected intracellularly to a similar level? Please discuss.

3) Discussion is lengthy. It should be shortened.

Minor comments:

There are some typo and grammatical errors, that need to be corrected.

Reviewer #2: PONE-D-19-20746

Characterization of Acp5 Missense Mutations Encoding Tartrate-Resistant Acid Phosphatases Associated with Spondyloenchondrodysplasia

This paper characterizes the Acp5 mutations encoding TRACP that commonly associated with Spondyloenchondrodysplasia (SPENCD). SPENCD is a recently identified disease, and it seems promising and exciting to investigate such disorder at the molecular level.

However, while reading the MS, I felt lost. Data is impressive. Clearer presentation is needed. The overall idea was not well-explained. MS needs rewriting. Some of the general and specific comments are stated below:

• General comments:

o The overall purpose of this study was not stated clearly.

o Excessive explanation in the result section. This makes some sorts of confusion. Every result subsection should plainly and simply state the result, whereas any other explanations should be carried out to the "discussion” section.

o There should be more introductions about Spondyloenchondrodysplasia and TRACP proteins in general. The relationship between the diseases and TRACP has to be elaborated in much detail.

o This reviewer is not clear about the bands in the Western analyses. MW markers are not provided. Several lanes are provided with no lane numbers.

o How do you normalize the secreted and intracellular proteins? Why do you think the top band as a non-specific band and consider that as a loading control? For secreted proteins in media, it is essential to show a Coomassie blue stained gel as a loading control. GAPDH or any other intracellular loading control should be added for normalization of the intracellular proteins

• Specific comments:

o The underlying reason for using HEK-293 cell line is not stated at all. Why not using other cell lines that not ubiquitously expressing TRACP?

o Figure 2: No clear illustration or indication of the stainable activity of cells. Quantification would also help.

o Result: the result section should correspond with the numerical order of the figures. The readers must not jump between figures while reading.

o Results: figure 6B does not correlate with what mentioned in the corresponding result section.

o Figures: Western analyses: comments provided under general comments

o Figures: Quantification alone for figure 7 is not sufficient. It is better to provide visual representations such as immunostaining and Western blotting analyses.

6. PLOS authors have the option to publish the peer review history of their article (what does this mean?). If published, this will include your full peer review and any attached files.

Reviewer #1: No

Reviewer #2: No

---

## [Author Response · Author response to Decision Letter 0]

5 Nov 2019

The authors profusely thank the reviewers’ for their constructive and valuable criticism which we strongly believe making our draft paper more refined and may be acceptable at this time. We earnestly apologize for typos, errors and delay in responding to review as our labs are being shifted from one campus to the other which made “our hands tied” to respond to all the recommendations. As the authors emanate from a non-English speaking country, we still learning and refining our way of writing. Hope we will get better in the future and certainly referees’ recommendations are tremendous helpful to refine our paper. This time we have given our manuscript to an English professor who was kind enough to correct the grammatical errors and typos. Figures which are ambiguous and “not scientifically talking back” are either removed or renumbered.

Again the authors thank the referees and we feel at this time, this revised manuscript should be acceptable in PLOSOne, which we consider as highest honor to publish from the east end. We will be happy to modify further if needed. Details on SPENCD disorder related abnormalities are coming up sharply in the last decade. 

Referee 1:

Authors profusely thank the Referee 1 for the constructive and immensely helpful criticisms.

1. We surely apologize for the conspicuous errors in the Results section for placing the figure numbers upside down. As recommended we now either correctly and sequentially place or change the figure numbers. Entire Result section was rewritten eliminating unwanted details so that both figures and results will coincide with each other. 

2. As rightly pointed out by the Referee, figure legends are rewritten so that they are devoid of experimental details. Experimental repeats were also mentioned.,

3. As recommended the figures 6a and 6b were renumbered as 6 and 7.

4. With regard to the observation of similar levels of intracellular TRACP mutant proteins.

We have discussed in appropriate section with regard to the similar levels of TRACP mutant proteins (Results and Discussion as well). First, we did all the experiments with stably transfected cell lines which took a large amount of time to perfect in between experiments. Realizing the cell number variations in different experiments may generate spurious values, we culture exact same number of cells per flask by counting both before and after culturing. This technique largely eliminated the quantitative variations between experiments, and we were able to do this as our personnel are fully trained and routinely do these techniques for hematology labs.

5. As recommended the discussion was sharply shortened and now it should be suffice.

6. Most of our typos and grammatical errors were corrected with the help of an English professor. As the authors are from a non-English speaking country we still evolving and hope umbrage will not be taken for this inability and deficiency. 

Referee 2:

1. Overall aim of the paper is now mentioned in the introduction.

2. As recommended excessive explanation in the Results section was eliminated in toto and discussion section was also modified and shortened.

3. Relationship between SPENCD and TRACP was also elaborated in the Introduction.

4. Molecular weight of TRACP isoforms 5a and 5b in the Western blots were indicated now. Putting together to make a meaningful picture was a herculean task for us and therefore we have used professional help to put together our multiple pictures (using Adobe Photoshop by Ms. Barbara J, please vide acknowledgement). We normally run standard proteins markers and to put together multiple runs at different points with standard marker proteins (which have small difference in migration and therefore they were removed). Now we marked the molecular weights of 5a and 5b proteins as they are the only primary bands of interest.

Thanking you.

---

## [Decision Letter · Decision Letter 1]

4 Dec 2019

PONE-D-19-20746R1

Characterisation of Acp5 Missense Mutations Encoding Tartrate-Resistant Acid Phosphatase Associated with Spondyloenchondrodysplasia

PLOS ONE

Dear Dr. Venugopal,

Thank you for submitting your manuscript to PLOS ONE. After careful consideration, we feel that it has merit but does not fully meet PLOS ONE’s publication criteria as it currently stands. Therefore, we invite you to submit a revised version of the manuscript that addresses the points raised during the review process.

ACADEMIC EDITOR: I suggest the authors to proof read the manuscript for English language and follow the journal format as noted in the reviewers comments below. Please address the  several minor comments raised by rev#3 carefully.

We would appreciate receiving your revised manuscript by Jan 18 2020 11:59PM. To enhance the reproducibility of your results, we recommend that if applicable you deposit your laboratory protocols in protocols.io, where a protocol can be assigned its own identifier (DOI) such that it can be cited independently in the future. For instructions see: http://journals.plos.org/plosone/s/submission-guidelines#loc-laboratory-protocols

We look forward to receiving your revised manuscript.

Kind regards,

Dr. Sakamuri V. Reddy

Academic Editor

PLOS ONE

Reviewers' comments:

Reviewer's Responses to Questions

**Comments to the Author**

1. If the authors have adequately addressed your comments raised in a previous round of review and you feel that this manuscript is now acceptable for publication, you may indicate that here to bypass the “Comments to the Author” section, enter your conflict of interest statement in the “Confidential to Editor” section, and submit your "Accept" recommendation.

Reviewer #1: (No Response)

Reviewer #2: All comments have been addressed

Reviewer #3: (No Response)

2. Is the manuscript technically sound, and do the data support the conclusions?

Reviewer #1: Partly

Reviewer #2: Yes

Reviewer #3: Yes

3. Has the statistical analysis been performed appropriately and rigorously? 

Reviewer #1: Yes

Reviewer #2: Yes

Reviewer #3: Yes

4. Have the authors made all data underlying the findings in their manuscript fully available?

Reviewer #1: No

Reviewer #2: Yes

Reviewer #3: Yes

5. Is the manuscript presented in an intelligible fashion and written in standard English?

Reviewer #1: No

Reviewer #2: Yes

Reviewer #3: No

6. Review Comments to the Author

Reviewer #1: There is only slight improvement in the revised manuscript.

Major issues:

1) Authors have not provided point-by-point response in detail. Very casual approach is taken by authors in writing of the response to reviewers comments.

2) There is only slight improvement in English writing in the manuscript.

3) However, the scientific language is still very poor throughout the manuscript, particularly the authors have not used proper scientific terms and framing of the sentences and grammar. It appears that manuscript is not properly edited by senior authors.

3) Result section is still very poor. Results of all the experiments are not written in details. The sub-figures are not cited in the text. Although, there are several sub-figures in many figures such as Figures 3, 4, 5 and 9, authors have just written there numbers.

4) Many Figures Legends are incomplete. e.g. In Figure 1 Legend, it is mentioned that missence mutations are shown below the WT sequence. This is not reflected in actual figure. Similarly, there are many mistakes in other Figures legends.

Reviewer #2: PONE-D-19-20746R1

Characterisation of Acp5 Missense Mutations Encoding Tartrate-Resistant Acid

Phosphatase Associated with Spondyloenchondrodysplasia

The MS has been revised as suggested, and the authors corrected the crucial issues.

It can be accepted for publication

Reviewer #3: Comments to the Author:

This article attempt to characterize the eight already reported missense mutations in Acp5 which is associated with spondyloenchondrodysplasia (SPENCD). The authors identified that the presence of mutants TRACP proteins are unable to induce enzymatic activity. Further, except T89I and M264K mutant proteins, others are in denatured precursor forms. The authors concluded that understanding the structural and functional aspects of mutant TRACP could lead to better understating of immune response and bone metabolism as well as targeting drug development.

The concept of the paper is relevant however there are many concerns that should be addressed before publication. The authors should follow the scientific writing format rather than common words and sentences. The whole manuscript should be corrected for details, clarity, and grammar.

Major issues:

1) In Abstract: RT-qPCR analysis is mentioned but there is no such figure and result in appropriate sections.

2) Methods section written poorly, refer the articles and follow the formats. For instance, western blotting is written without clarity and details. Some of the Methods written like results that should be avoided.

3) It is not appropriate to write a sentence like this in the result section “Other salient features of TRACP are described in Figure 1 legend”. Provide more details about TRACP enzyme in this section.

4) “The steady state transcription ….10-fold lower compared to WT”. There is no corresponding figures/results in this manuscript.

Minor issues:

1) In Abstract: At the end, summarize or conclude the work and it should exactly reflect the results. Then write what it could be useful to, such as drug development and underlying mechanisms, etc.

2) In Abstract: “Certainly, determining the structure functional relationships in TRACP will expand….” Should modify to “Determining the structural and functional relationships of TRACP may expand… “

3) An unscientific sentence like this and many others throughout the manuscript should be modified “TRACP has many functions in vitro and its biological role in bone resorption and immune responses is becoming clearer”. What exactly authors want to convey with this sentence “TRACP has many functions in vitro….”

4) “Therefore, serum TRACP5a may serve as a marker for systemic macrophage number…” is it possible to count the macrophage number using serum TRACP5a level? Similarly, isoform 5b release and osteoclast number? Clarify both the statements.

5) Mentioned in this sentence that “We and others have shown earlier [9, 10] that mutations…” but there is one other’s paper cited, add more reference or modify.

6) Follow the same format for gene name, ACP5 or Acp5.

7) Add references to these sentences and many others "More specifically, ….TRACP enzyme”, “Other salient features…..calcification”

8) “Transfection of synthetic genes” This subtitle should be written like this “Transformation of mutant genes and subcloning” and describe the methods in the text.

9) “Transient transfection and generation of stable cell line” This should be a subtitle for the next paragraph (from this sentence onwards “HEK-293 cells were transfected with……recommended procedures.”

10) What is the loading control used for western blot analysis? Explain.

11) Figures should be labeled appropriately. For instance, Fig 2, TRAP staining and Mab220.

12) Y-axis scale difference is high between min and max in many bar graphs, Therefore if it appropriate to represent the graph with the broken y-axis to show the lower values.

13) The results section should be written clearly with details.

7. PLOS authors have the option to publish the peer review history of their article (what does this mean?). If published, this will include your full peer review and any attached files.

Reviewer #1: No

Reviewer #2: No

Reviewer #3: No

---

## [Author Response · Author response to Decision Letter 1]

13 Feb 2020

PONE-D-19-20746R1

The authors profusely thank the reviewers for their constructive and valuable criticism which we strongly believe has enhanced the quality of the revised paper. We apologize for all errors in the earlier version. As recommended, we have now revised our manuscript according to the reviewers’ suggestions and we have tried our best by giving detailed responses to each of the reviewers’ questions point-by-point.

Reviewer #1: There is only slight improvement in the revised manuscript.

Major issues:

1) Authors have not provided point-by-point response in detail. Very casual approach is taken by authors in writing of the response to reviewers’ comments.

Reply to the Reviewer 1: We apologize for the conspicuous errors in the Results section. We have added more information and rewritten the results section in better English. 

2) There is only slight improvement in English writing in the manuscript.

Reply to the Reviewer 1: As recommended, we have now corrected the entire manuscript especially with regard to English and grammar.

3) However, the scientific language is still very poor throughout the manuscript, particularly the authors have not used proper scientific terms and framing of the sentences and grammar. It appears that manuscript is not properly edited by senior authors.

Reply to the Reviewer 1: As recommended several changes have been made throughout the manuscript with proper scientific terms and framing the sentences in better English.

3) Result section is still very poor. Results of all the experiments are not written in details. The sub-figures are not cited in the text. Although, there are several sub-figures in many figures such as Figures 3, 4, 5 and 9, authors have just written there numbers.

Reply to the Reviewer 1: Results section has been drastically modified and details added wherever necessary. As recommended, Discussion was also drastically shortened. The sub-figures have been cited in the text. We have revised the figure numbers and attached as individual files as recommended.

4) Many Figures Legends are incomplete. e.g. In Figure 1 Legend, it is mentioned that missense mutations are shown below the WT sequences. This is not reflected in actual figure. Similarly, there are many mistakes in other Figures legends.

Reply to the Reviewer 1: We have revised and rewritten the figure legends as recommended.

Figure 1 was carefully revised to reflect the details of TRACP. 

Reviewer #2: 

Characterization of Acp5 Missense Mutations Encoding Tartrate-Resistant Acid

Phosphatase Associated with Spondyloenchondrodysplasia The MS has been revised as suggested, and the authors corrected the crucial issues. It can be accepted for publication

Reply to the Reviewer 2: We thank you for your suggestions.

Reviewer #3: 

This article attempts to characterize the eight already reported missense mutations in Acp5 which is associated with spondyloenchondrodysplasia (SPENCD). The authors identified that the presence of mutants TRACP proteins are unable to induce enzymatic activity. Further, except T89I and M264K mutant proteins, others are in denatured precursor forms. The authors concluded that understanding the structural and functional aspects of mutant TRACP could lead to better understating of immune response and bone metabolism as well as targeting drug development. The concept of the paper is relevant however there are many concerns that should be addressed before publication. The authors should follow the scientific writing format rather than common words and sentences. The whole manuscript should be corrected for details, clarity, and grammar.

Major issues:

 1) In Abstract: RT-qPCR analysis is mentioned but there is no such figure and 

result in appropriate sections.

Reply to the Reviewer 3: Since the contribution by RT-qPCR study is minimal we have removed this section from our paper and changed all sections accordingly.

2) Methods section written poorly, refer the articles and follow the formats. For 

instance, Western blotting is written without clarity and details. Some of the Methods 

written like results that should be avoided.

Reply to the Reviewer 3: As recommended we have revised our Methods section and added more details to Western blot technique. Results reported in Methods section have been removed as per suggestion. 

3) It is not appropriate to write a sentence like this in the result section “Other salient features of TRACP are described in Figure 1 legend”. Provide more details about TRACP enzyme in this section.

Reply to the Reviewer 3: As suggested, we have provided details about TRACP enzyme in this Results section with Figure 1.

4) “The steady state transcription…..10 fold lower compared to WT”. Thereis no corresponding figures/results in this manuscript.

Reply to the Reviewer 3: As RT-qPCR has been removed in this revised version, sentences related to this work have also been removed from other sections of this resubmission. 

Minor issues:

1) In Abstract: At the end, summarize or conclude the work and it should exactly reflect the results. Then write what it could be useful to, such as drug development and underlying mechanisms, etc.

Reply to the Reviewer 3: As recommended, we have completely rewritten the Abstract..

2) In Abstract: “Certainly, determining the structure functional relationships in TRACP will expand….” Should modify to “Determining the structural and functional relationships of TRACP may expand… “

Reply to the Reviewer 3: As recommended, we have modified these sentences as “Determining the structural and functional relationships of TRACP may expand…”.

3) An unscientific sentence like this and many others throughout the manuscript should be modified “TRACP has many functions in vitro and its biological role in bone resorption and immune responses is becoming clearer”. What exactly authors

want to convey with this sentence “TRACP has many functions in vitro….”

Reply to the Reviewer 3: As recommended, we have modified this sentence. Please see our revised manuscript.

4) “Therefore, serum TRACP5 may serve as a marker for systemic macrophage number….” Is it possible to count the macrophage number using serum TRACP5a level? Similarly, isoform 5b release and osteoclast number?? Clarify both statements.

Reply o the Reviewer 3: We have modified the sentence with clarity in this revised version.

5) Mentioned in this sentence that “We and others have shown earlier [9, 10] that mutations…” but there is one other’s paper cited, add more reference or modify.

Reply to the Reviewer 3: As recommended, we have now added more references here.

6) Follow the same format for gene name, ACP5 or Acp5.

Reply to the Reviewer 3: We apologize for this inconsistency. We have consistently followed the same pattern of human gene nomenclature. Human gene should be in caps and we have changed accordingly throughout the manuscript.

7) Add references to these sentences and many others "More specifically, ….TRACP enzyme”, “Other salient features…. calcification”

Reply to the Reviewer 3: As suggested, we have now added references to these sentences.

8) “Transfection of synthetic genes” This subtitle should be written like this “Transformation of mutant genes and subcloning” and describe the methods in the text.

Reply to the Reviewer 3: As directed, the subtitle has been changed accordingly.

9) “Transient transfection and generation of stable cell line” This should be a subtitle for the next paragraph (from this sentence onwards “HEK-293 cells were transfected with……recommended procedures.”

Reply to the Reviewer 3: As suggested, the subtitle has been inserted in its place.

10) What is the loading control used for Western blot analysis? Explain

Reply to the Reviewer 3: Since there are no other proteins present in the blot other than our electrophoretically homogeneous TRACP proteins, we treat this protein as loading control. Also the electrophoresis combs are not sufficient enough to increase the number of samples. 

11) Figures should be labeled appropriately. For instance, Fig 2, TRAP staining and Mab220.

Reply to the Reviewer 3: We thank the reviewer for the comments. As rightly pointed out by the Referee, figures have been rewritten appropriately, and the figure legends have also been rewritten so that they are devoid of experimental details. 

12) Y-axis scale difference is high between min and max in many bar graphs, therefore if it appropriate to represent the graph with the broken y-axis to show the lower values.

Reply to the Reviewer 3: We have revised the graphs as much as the Software package allows. 

13) The results section should be written clearly with details.

Reply to the Reviewer 3: We have corrected the Results section with clarity now.

---

## [Decision Letter · Decision Letter 2]

21 Feb 2020

Characterisation of Acp5 Missense Mutations Encoding Tartrate-Resistant Acid Phosphatase Associated with Spondyloenchondrodysplasia

PONE-D-19-20746R2

Dear Dr. Venugopal,

We are pleased to inform you that your manuscript has been judged scientifically suitable for publication and will be formally accepted for publication once it complies with all outstanding technical requirements.

With kind regards,

Sakamuri V. Reddy, Ph.D

Academic Editor

PLOS ONE

Additional Editor Comments (optional):

Reviewers' comments:

Reviewer's Responses to Questions

**Comments to the Author**

1. If the authors have adequately addressed your comments raised in a previous round of review and you feel that this manuscript is now acceptable for publication, you may indicate that here to bypass the “Comments to the Author” section, enter your conflict of interest statement in the “Confidential to Editor” section, and submit your "Accept" recommendation.

Reviewer #1: All comments have been addressed

Reviewer #3: All comments have been addressed

2. Is the manuscript technically sound, and do the data support the conclusions?

Reviewer #1: Yes

Reviewer #3: Yes

3. Has the statistical analysis been performed appropriately and rigorously? 

Reviewer #1: Yes

Reviewer #3: Yes

4. Have the authors made all data underlying the findings in their manuscript fully available?

Reviewer #1: Yes

Reviewer #3: Yes

5. Is the manuscript presented in an intelligible fashion and written in standard English?

Reviewer #1: Yes

Reviewer #3: Yes

6. Review Comments to the Author

Reviewer #1: Sub figures such as A, B, C etc.of Figures 4, 5 are not yet cited in Resullts. That need to be corrected.

Reviewer #3: (No Response)

7. PLOS authors have the option to publish the peer review history of their article (what does this mean?). If published, this will include your full peer review and any attached files.

Reviewer #1: No

Reviewer #3: No

---

## [Editor Report · Acceptance letter]

9 Mar 2020

PONE-D-19-20746R2 

Characterisation of *Acp5* Missense Mutations Encoding Tartrate-Resistant Acid Phosphatase Associated with Spondyloenchondrodysplasia 

Dear Dr. Venugopal:

I am pleased to inform you that your manuscript has been deemed suitable for publication in PLOS ONE. Congratulations! Your manuscript is now with our production department. 

With kind regards,

on behalf of

Dr. Sakamuri V. Reddy 

Academic Editor

PLOS ONE